# Design, Synthesis and Pharmacological Evaluation of Novel C^2^,C^3^-Quinoxaline Derivatives as Promising Anxiolytic Agents

**DOI:** 10.3390/ijms232214401

**Published:** 2022-11-19

**Authors:** Dmitriy V. Maltsev, Maria O. Skripka, Alexander A. Spasov, Pavel M. Vassiliev, Maxim A. Perfiliev, Lyudmila N. Divaeva, Alexander A. Zubenko, Anatolii S. Morkovnik, Alexander I. Klimenko, Mikhail V. Miroshnikov, Vladlen G. Klochkov, Laura R. Ianalieva

**Affiliations:** 1Department of Pharmacology and Bioinformatics, Volgograd State Medical University, 1 Pavshikh Bortsov sq., 400131 Volgograd, Russia; 2Volgograd Medical Research Center, 1 Pavshikh Bortsov sq., 400131 Volgograd, Russia; 3Research Institute of Physical and Organic Chemistry, Southern Federal University, 105/42 Bolshaya Sadovaya Str., 344090 Rostov-on-Don, Russia; 4North-Caucasian Zonal Research Veterinary Institute, 346406 Novocherkassk, Russia

**Keywords:** quinoxaline, anxiolytic, pharmacophore, diazepam, ADMET

## Abstract

A new series of quinoxaline derivatives, **2a**–**4b**, were synthesized and their anxiolytic potential was evaluated in vivo using elevated plus maze (EPM), open field (OF) and light-dark box (LDB) techniques. According to the results of the EPM, four active compounds were found in **2a**, **2b**, **2c**, **4b**. Their anxiolytic properties were confirmed in terms of LDB and the most active was compound **2b**. In the OF, only **2c** had an influence on the locomotor activity of the rodents. Thus, the most promising substance was determined; this was **2b**, which has the structure of 2-(2-{[3-(4-tert-butylphenyl)quinoxaline-2-yl]methyl}-4,5-dimethoxyphenyl)-N-methylethan-1-amine hydrochloride. The obtained data were analyzed with the pharmacophore feature prediction approach, which made it possible to compare the structures of the studied compounds with the reference drug diazepam, and to determine the contribution of pharmacophores to the manifestation of the activity under study. ADMET analysis was carried out for compound **2b** and the acute oral toxicity of this substance was also tested in vivo. As a result of the study, a promising compound with a high anxiolytic effect and low level of toxicity **2b** was found, which is of interest for further preclinical study of its properties.

## 1. Introduction

Mental illnesses are estimated to affect more than one billion people worldwide [1,2]. On average, one in three individuals will suffer from a mental disorder during their lifetime [3]. More so than any other type of disease, mental disorders are subject to prejudices, marginalization and stigmatization [4]. In addition to epidemiological measures, such as morbidity and mortality, the importance of mental disorders can also be investigated through cost-of-illness studies within health economics: high economic burden caused by the global direct and indirect economic costs of mental disorders are estimated to be 2.5 trillion USD [5].

Anxiety disorders are the most common mental illnesses [6]. As a normal emotion, anxiety performs an adaptive function under stressful circumstances, but may become inadequate when it does has not any motivation and, in this case, constitutes a clinical syndrome [7]. During the epidemic of COVID-19, the prevalence of anxiety states increased, as well as depression and anger, while positive emotions and life satisfaction decreased [8]. Many suffer from uncertainty, fear of infection, moral distress and grief, often experienced alone [9].

At present, selective serotonin reuptake inhibitors and selective serotonin norepinephrine reuptake inhibitors are recommended as first-line treatments due to their good benefit/risk balance. On the other hand, they still have significant side effects such as sexual dysfunction, serotonin syndrome, withdrawal symptoms and dependence generation [10,11,12]. Current guidelines also do not recommend benzodiazepines as first-line treatments as they cause a number of undesirable effects, including unwanted sedation and impaired concentration [13]. Anxiolytic medication shows drug activity to a pool of biological targets, sometimes non-selectively to a series of them, which can lead to a decrease in therapeutic efficacy, with an increased level of toxicity due to reactive metabolites. Therefore, the first step in drug development and design is to assess risks, predict the likeness of substances, and establish QSAR using the in silico approach. The search for novel compounds with anxiolytic properties and less pronounced adverse reactions among a wide range of chemical compounds continues to the present [14,15,16,17]. The anxiolytic potential of quinoxaline has been described earlier in the literature sources [18,19,20,21]. Thus, the present study is devoted to the synthesis and pharmacological evaluation of a new series of quinoxaline derivatives.

Heteroaryl-substituted derivatives of monoamine neurotransmitters, neuromodulators and neurohormones, such as dopamine, adrenaline, noradrenaline, serotonin, melatonin, histamine and β-phenylethylamine, as well as their close structural analogues, have attracted considerable attention from researchers (see for example: [22,23,24,25,26,27,28,29,30,31]). Due to this relationship, these compounds should be of interest as neurotropic agents.

In this study, we synthesized the previously unknown type of heteroarylated analogues of monoamine neurotransmitters, namely, o-quinoxalyl-2-methyl derivatives of β-arylethylamines and 2-quinoxalyl-2-methyl N-methyltryptamine. The suggested synthetic approach is based on the recyclization reactions of 4-acyl-1,2,3,4-tetrahydrobezo[d]azepines and their dihydrocarboline analogues with o-phenylenediamine as a recyclizating agent [32]. In this work, we have obtained a number of new compounds of this type and studied the effects of such quinoxalyl-methyl derivatives on the central nervous system of laboratory animals. The synthesis routes are illustrated by Figure 1.

## 2. Results

### 2.1. Chemistry

The solvents were purified according to standard procedures. NMR spectra were recorded at 30°C on a Bruker Avance 600 (600 MHz) spectrometer in DMSO-*d*_6_. Chemical shifts of nuclei ^1^H and ^13^C were measured relative to the reference peaks of the signals of deuterated solvent: [δ = 2.50 ppm for residual protons and 39.09 ppm for carbon [33]. Coupling constants (*J*) are reported in Hz. Melting points were determined by using Fisher-Johns Melting Point Apparatus (Fisher Scientific) and are uncorrected. Elemental analysis was performed by the traditional method of microanalysis. The reaction and purity of the obtained compounds were monitored by TLC (plates with Al_2_O_3_ III activity grade, eluent CHCl_3_, development of TLC plates by exposition to iodine vapors in “iodine chamber”).

The starting condensated acyldihydroazepines **1** were provided by InterBioscreen Ltd. (Russia). Compounds **2a**, **2c**, **2e**, **4a** were synthesized as described previously [32]. The structure of the studied C^2^,C^3^-quinoxaline derivatives are presented in Table 1 (see also Appendix A).

### 2.2. Behavioral Tests

#### 2.2.1. Elevated plus Maze

On average, the experimental mice spent 6.8 ± 1.2% of the time in the light arms of EPM in the control group and 22.0 ± 1.6% in the diazepam group. The tested compounds manifested various levels of anxiolytic activity. Under the action of compound **2d**, the mice spent even less time in the illuminated areas than in the control group: only 0.5 ± 0.3%. The level of the effect of compounds **2e**, **2f**, **2g**, **4a** corresponded to the control values. The behavior of the mice under the influence of substances **2a** and **4b** was similar to the group of the comparison drug. Compound **2c** also showed a significant reduction in stress behavior compared to the control group, although this was slightly below the level of diazepam and substances **2a** and **4b**. The highest level of effect in this case was registered for the compound under code **2b**–28.2 ± 2.0%. According to the parameter of the number of entries of rodents into the open arms of EPM, a significant result was noted only under the influence of diazepam and compounds **2a** (*p* < 0.05) and **2b** (*p* < 0.01) (Figure 1).

#### 2.2.2. Open Field

The number of squares crossed by rodents of the control group was approximately 73.3 ± 4.4, while in the diazepam group it was 83.5 ± 10.2. The parameter did not change significantly in all tested groups, with the exception of **2c**, which was 127.7 ± 12.7 squares on average. There were also no significant changes in the rearings score, which is evidence of an absence of the muscle relaxant effects of the compounds under study. The number of holes examined by the mice (search activity) was higher in the rodents treated with diazepam (9.1 ± 2.0) than in the controls (3.6 ± 0.8); however, a significant difference was not registered. Under the influence of compound **2b,** an increase in search activity was observed (8.6 ± 1.0), up to the level of diazepam, but also without significance when compared to the control values. In terms of the number of entries to the center of the open field, significant changes compared to the control group (0.6 ± 0.3) were registered in the mice treated with diazepam (4.3 ± 0.7) and compounds **2a** (4.8 ± 0.9), **2b** (4.8 ± 0.8), **2c** (4.1 ± 0.7) and **4b** (4.8 ± 0.9). The values for long and short self-grooming acts, which can be indirect parameters of calm and nervous conditions of animals, respectively, did not differ sufficiently to form conclusions (Figure 2).

#### 2.2.3. Light/Dark Box

The light/dark box was used for testing the most active substances due to the results of the elevated plus maze and open field techniques. The mice administered with the vehicle spent 14.1 ± 3.4% of the observation time in the light chamber. This value increased sufficiently in the groups of the comparison drug (45.5 ± 3.7%) and compounds **2a** (45.0 ± 4.4%), **2b** (53.7 ± 4.1%), **2c** (48.0 ± 3.6%) and **4b** (48.5 ± 2.0%). Thus, all compounds under study with the highest values of anxiolytic activity confirmed their properties in terms of the light/dark box technique (Figure 3).

### 2.3. In Silico

#### 2.3.1. Pharmacophore Feature Prediction

The data sets for predictions are divided into three main groups: the group with diazepam set1, with diazepam as a key molecule for pharmacophore prediction; the second set2, with NBQX as a key molecule; and the third group set3 without diazepam and NBQX, to evaluate the pharmacophore set for quinoxalines.

The set1 group is characterized by only three variants of the calculated pharmacophore groups, which have three features; for all of them, the presence of an aromatic group, one or two, is specific, as well as hydrophobic and acceptor features (Table 2). The second set2 group does not differ from the first; only for the group where the key ligand was NBQX, for which already four features, without changes in the qualitative composition of the features. The scoring function is ranges between 14.230 and 27.042, which corresponds to group set1 (Table 3). The third group was critically different from the first two; it is characterized by six to eight features for the complete alignment of all nine quinoxoline molecules. The score function for this set ranged between 42.608 and 48.454, which exceeds the same data for the set1 and set2 sets by almost two times. The best results, in terms of the number of features, as well as the score function, were obtained for the data set in which the key molecules were **2b**, which show a fairly high activity in the experiment (Table 4).

The three-dimensional coordinates and characteristics analysis obtained during the prediction is visualized using UCSF ChimeraX, LigandScout 4.4 and Discovery Studio Visualizer V21.1.0.20298.

For the first set of Figure 4, it can be seen that the overlapping of objects did not occur in the best way, but relative to the diazepam molecule. Despite the apparent large number of pharmacophore groups, they are ungrouped and do not have common areas. The set2 dataset has a more pronounced centrality, but this did not affect the number of features in any way. The best result was in the set3 group, shown in Figure 5; this set had excellent alignment and the greatest number of features.

Figure 5 provides a more detailed look at the alignment and common pharmacophore groups specific to the datasets. LigandScout confirms the previous alignments for each group: the same pharmacophore fragmentation for the first group in set1, similar motifs for the second set2, as well as full agreement for the diazepam-free data set, are maintained. However, there are different types of pharmacophores within the three sets. There were two hydrophobic interactions for set1 and set3; five for set2; a hydrophobic bond acceptor in sets one, two and four, respectively; and one unique hydrophobic bond donor group for set3.

#### 2.3.2. ADMET Properties

We can predict that compound **2b** will absorb GI well but permeate BBB poorly. This must be confirmed in further pharmacokinetic studies due to the high substance activity, which must be connected with the action on CNS. The fact that there were no alerts in the PAINS and Bank checks is a good further development sign for the drug. Four class of acute oral toxicity was predicted (Table 5).

### 2.4. Acute Oral Toxicity of Leader Compounds

As the 2000 mg/kg dose caused 100% animal mortality within 24 h, the decision was made to reduce the dose to 1000 mg/kg. When it was found that the introduction of this dose did not cause mortality in the experimental animals during the observation period, a search was made for the average lethal dose between the indicated values. The LD50 was then calculated using linear regression and was determined at the level of 1539.6 mg/kg. These data allow us to classify compound **2b** to the IV class of toxicity, being practically non-toxic (Figure 6).

## 3. Discussion

Fear is a fundamental emotion that is necessary for survival [34]. In turn, anxiety arises without an adequate triggering stimulus, and is an affective state in which a person prepares himself for uncertain, but possible negative outcomes. Anxiety, the symptoms of which include irritability, tension and inability to relax, a feeling of “nervousness”, memory impairment, and sleep disturbance [35] is also characterized by a number of somatic manifestations such as chronic pain caused by muscle hypertension, cardiovascular, and even pseudo-allergic symptoms [36,37]. It has also a high comorbidity with other syndromes. Although anxiety and depression have been considered as two distinct entities, according to the diagnostic criteria, anxious depression is a relatively common syndrome [38]. The link between mood disorders and acute pain has proven to be increasingly significant as the link is bi-directional, and both act as risk factors for the other [39]. In some individuals, anxiety disorders can even precede the development of motor symptoms of Parkinson’s disease, caused by neurodegeneration [40].

Psychological treatments of anxiety disorders, including cognitive behavior therapy, stress management, mindfulness training, and acceptance and commitment therapy, given either individually, in groups, or over the internet, have all proved efficacious in both the short and longer term [41]. At the same time, psychotherapy, in association with pharmacotherapy, is associated with better efficacy [13], and thus pharmacological treatment continues to evolve. Searching for novel compounds with anxiolytic properties is a developing field in medicinal chemistry and pharmacological preclinical studies and is conducted among a wide range of derivatives of a various chemical classes including piperazine, pyrazoloquinoline, β-phenylglutamic acid, azepinobenzimidazole, benzotriazine, N-propylcinnoline, imidazopyridine, triazine, benzoxazole, arylsulfonamide, synthetic flavonoids and many others [15].

As mentioned earlier, we synthesized the novel type of heteroarylated analogues of monoamine neurotransmitters, and thus these derivatives are of interest for studying their neurotropic activity. Whilst the study of anxiolytic potential of quinoxaline derivatives is ongoing [18,19], our compounds differ significantly from them in molecule structure and are entirely new. 

The study began with estimating the in vivo anxiolytic activity of the number of C^2^,C^3^-quinoxaline derivatives with the standard methods: elevated plus maze and open field. The most active substances, according to the results of these tests, were **2a**, **2b**, **2c** and **4b**, and these were additionally tested using the light/dark box technique. The compound with the highest activity was chosen according to the totality of the results of in vivo tests, and this was compound **2b**. The animals under the influence of this compound spent significantly more time in the illuminated compartments of the installations, showed interest in exploring new spaces, behaved calmly, displayed no signs of muscle relaxation.

The in silico prediction data showed the prospect of studying quinoxalines as a reference set in the search for active anxiolytic compounds, however, for a detailed and more accurate prediction, an extended data set is required, where a reference drug of a similar chemical structure or a group of compounds with a reliable level of activity will serve as the basis for the pool of pharmacophore candidates’ outcomes.

The PharmGist pharmacophore prediction best results show three features in set1: two aromatics and one acceptor, and it gained ten molecules with diazepam. Set2 had a similar score and almost the same number of features: one aromatic and three acceptors. However, set3 had features two times higher: three aromatic, three acceptors, one hydrophobic and one donor, meaning that not only do the number of aromatic or acceptors play a crucial role in the pharmacophore prediction, but the type of the features do too. Although set2, with NBQX, had the same scaffold as the all new quinoxalines, this did not define high scores for the in silico test; thus, the dataset without significant structural differences show the best results. Among the substances, the most promising compound, according to the prediction, is **2b,** which showed the best score and acted as the key compound of the group.

The level of acute oral toxicity in the mice was relatively low, allowing us to classify compound **2b** as practically non-toxic (IV class of toxicity). Thus, further investigating the properties of **2b** is promising due to its low toxicity and high anxiolytic effect compared to the reference drug, diazepam.

## 4. Materials and Methods

### 4.1. General Procedure for the Synthesis of Compounds ***2b***, ***2f***, ***2g***, ***4b***

A mixture of the 3 mmol acylated dihydroazepine **1b**, **1f**, **1g** or **3b**, 3 mmol (0.32 g) o-phenylenediamine, 1 mL conc. HCl in 10 mL EtOH was refluxed for 2 h and a solution of 1.0 g Na_2_CO_3_ in 20 mL water added. Then, product **4b** was filtered and washed with water (4 × 3 mL), whereas amines **2b**, **2f**, **2g** were extracted with chloroform (2 × 10 mL). The chloroform extract was dried with anhydrous Na_2_CO_3_ and the solvent was evaporated to dryness, in vacuo, at 50–60 °C. Subsequently, the compounds **2b**, **2f**, **2g**, **4b** were purified by column chromatography (Al_2_O_3_, CHCl_3_) and converted to hydrohalids by treatment with mixture of conc. HCl (for **2b**, **2f**, **2g, 4b**) or HBr (for **2d**) in EtOH. The final purification of the hydrohalids was carried out by recrystallization from EtOH (for **2b**, **2d**, **2f**, **2g**) or PrOH (for **4b**). The structures of hydrochlorides C^2^,C^3^-quinoxaline derivatives **2b**, **2f**, **2g**, **4b** and hydrobromide **2d** are shown in Figure 7, Figure 8, Figure 9, Figure 10 and Figure 11.

#### 4.1.1. 2-(2-{[3-(4-(Tert-butyl)phenyl]quinoxalin-2-yl]methyl}-4,5-dimethoxyphenyl)-N-methylethanamine hydrochloride (**2b**)

The starting compound was (4-(tert-butyl)phenyl)(7,8-dimethoxy-3-methyl-2,3-dihydro-1H-benzo[d]azepin-4-yl)methanone **1b**. The yield of **2b** was 67%. Colorless crystals with mp 141–143 °C (from EtOH). ^1^H NMR (600 MHz), DMSO-*d*_6_. δ, ppm: 1.34 (s, 9H, *^t^*Bu), 2.46–2.49 (m, 3H, NMe), 2.90–2.99 m, 2H, H-α), 3.00–3.04 (m, 2H, H-β), 3.44 (s, 3H, OMe), 3.71 (s, 3H, OMe), 4.43 (s, 2H, ArCH_2_Het), 6.30 (s, 1H, H-6′), 6.81 (s, 1H, H-3′), 7.53–7.57 (m, 2H, H-3″, H-5″), 7.67–7.71 (m, 2H, H-6, H-7), 7.78–7.82 (m, 2H, H-2″, H-6″), 7.98–8.02, (m, 1H, H-5), 8.03–8.07 (m, 1H, H-8), 8.98–9.08 (m, 2H, +NH_2_). ^13^C NMR (150 MHz), DMSO-*d*_6_. δ, ppm: 28.47, 31.04, 32.21, 34.47, 38.48, 48.89, 55.31, 55.49, 113.48, 113.73, 125.12, 128.10, 128.28, 128.70, 128.74, 129.03, 129.72, 129.94, 135.83, 140.18, 140.33, 147.21, 147.34, 151.47, 154.58, 154.71. Found (%): C, 71.03; H, 7.00; Cl, 7.22, N, 8.03. Calc. for C_30_H_36_ClN_3_O_2_ (%): C, 71.20; H, 7.17; Cl, 7.01, N, 8.30.

#### 4.1.2. 2-(2-{[3-(3,4-Dichlorophenyl)quinoxalin-2-yl]methyl}-4,5-dimethoxyphenyl)-N-methylethanamine hydrobromide (**2d**)

The starting compound was 2-(2-{[3-(3,4-dichlorophenyl)quinoxalin-2-yl]methyl}-4,5-dimethoxyphenyl)-N-methylethan-1-amine **1d**. It was converted to hydrobromides by treatment with mixture of conc. HBr and EtOH. The yield was 89%. Colorless crystals with mp 186–187 °C (from EtOH). ^1^H NMR (600 MHz), DMSO-*d*_6_. δ, ppm: 2.51 (s, 3H, NMe), 2.78–2.75 (m, 2H, H-α), 2.97–3.08 (m, 2H, H-β), 3.51 (s, 3H, OMe), 3.73 (s, 3H, OMe), 4.39 (s, 2H, ArCH_2_Het), 6.38 (s, 1H, H-6′), 6.81 (s, 1H, H-3′), 7.71 (dd, *J* 8.2, 2.1, 1H, H-3″), 7.79 (d, *J* 8.2, 1H, H-2″), 7.82–7.87 (m, 2H, H-6, H-7), 7.90 (d, *J* 2.0, 1H, H-6″), 8.01–8.05 (m, 1H, H-8), 8.07–8.13 (m, 1H, H-5), 8.47 (s, 1H, ^+^NH_2_). ^13^C NMR (150 MHz), DMSO-*d*_6_. δ, ppm: 28.36 (C-CH_2_-group of fragment CH_2_-CH_2_), 32.32 (NMe), 38.63 (ArCH_2_Het), 48.75 (N-CH2), 55.26 (OMe), 55.52 (OMe), 113.45 (C-6′), 113.93 (C-3′), 127.85, 128.26, 128.28, (C-3″,5″), 128.81 (C-5′), 129.39, 129.96, 130.50, 130.98, 131.12, 131.82, 139.07, 139.93, 140.61, 147.28, 147.43, 152.37, 154.19. Found (%): C 55.21; H, 4.25; Br + Cl, 26.48; N, 7.31. Calc. for C_26_H_26_BrCl_2_N_3_O_2_ (%): C 55.44; H, 4.65; Br, 14.18; Cl, 12.59; N, 7.46.

#### 4.1.3. 1-(4,5-Dimethoxy-2-{[3-methylquinoxalin-2-yl]methyl}phenyl)-N,2-dimethylpropan-2-amine hydrochloride (**2f**)

The starting compound was 1-(2,2,3-trimethyl-2,3-dihydro-1H-benzo[*d*]azepin-4-yl)ethanone **1f**. The yield was 73%. Colorless crystals with mp 173–175 °C (from EtOH). ^1^H NMR (600 MHz), DMSO-*d*_6_. δ, ppm: 1.21 (s, 6H, 2Me), 2.45–2.47 (m, 5H, NMe, DMSO-*d*6), 2.66 (s, 3H, Me), 3.10 (s, 2H, 2H-α), 3.50 (s, 3H, OMe), 3.73 (s, 3H, OMe), 4.40 (s, 2H, ArCH_2_Het), 6.58 (s, 1H, H-6′), 6.85 (s, 1H, H-3′), 7.66–7.72 (m, 2H, H-3, H-5), 7.88 (dd, 1H, *J* 8.0, 1.7, H-7), 7.95 (dd, 1H, *J* 8.0, 1.7, H-4), 9.29–9.32 (m, 2H, ^+^NH_2_). ^13^C NMR (150 MHz), DMSO-*d*_6_. δ, ppm: 18.53, 22.08 (CH_2_ of fragment CH_2_CMe_2_), 22.08 (Me), 22.52 (Me), 25.82 (Me), 38.52 (ArCH_2_Het), 39.03 (NCH_2_), 55.35 (OMe), 55.26 (OMe), 59.49 (NMe), 113.82 (C-6′), 115.32 (C-3′), 126.51, 127.78, 128.35, 128.99, 129.04, 129., 140.01, 140.20, 146.96, 147.55, 153.87, 155.75. Found (%): C, 66.30; H, 7.00; Cl, 8.32, N, 10.42. Calc. for C_23_H_30_ClN_3_O_2_ (%): C, 66.41; H, 7.27; Cl, 8.52, N, 10.10.

#### 4.1.4. 1-(2-{[3-(4-Bromophenyl)quinoxalin-2-y])methyl}phenyl)-N,2-dimethylpropan-2-amine hydrochloride (**2g**)

The starting compound was 1-(2,2,3-trimethyl-2,3-dihydro-1H-benzo[*d*]azepin-4-yl)ethanone **1a**. The yield was 73%. Colorless crystals with mp 143–146 °C (from EtOH). ^1^H NMR (600 MHz), DMSO-*d*_6_. δ, ppm: 1.14 (s, 6H, 2Me), 2.47 (d, *J* 5.5, 3H, NMe), 2.93 (s, 2H, H-α), 4.51 (s, 2H, ArCH_2_Het), 6.74 (dd, *J* 7.7, 1H, H-3′), 7.04–7.07 (m, 1H, H-4′), 7.11–7.16 (m, 2H, H-5′, H-6′), 7.56–7.60 (m, 2H, H-3″, H-5″), 7.63–7.66 (m, 2H, H-6, H-7), 7.82–7.84 (m, 2H, H-5, H-8), 7.98–8.01 (m, 1H, H-2″), 8.06–8.10, (m, 1H, H-6″), 9.39–9.42 (m, 2H, +NH_2_). ^13^C NMR (150 MHz), DMSO-*d*_6_. δ, ppm: 18.50, 21.90 (CH_2_ of fragment CH_2_CMe_2_), 25.67 (2Me), 38.94 (ArCH_2_Het), 39.27 (NCH_2_), 59.22 (NMe), 122.39, 126.11, 126.99, 128.38, 128.77, 129.81, 129.94, 130.27, 131.05, 131.16, 131.56, 134.19, 137.49, 137.90, 140.12, 140.56, 153.87, 154.39. Found (%): C, 62.53; H, 5.22; Br + Cl, 23.00, N, 10.42. Calc. for C_26_H_27_BrClN_3_ (%): C, 62.85; H, 5.48; Br, 16.08; Cl, 7.14, N, 8.46.

#### 4.1.5. 2-(2-{[3-(3,4-Dichlorophenyl)quinoxalin-2-yl]methyl}-1H-indol-3-yl)-N-methylethan-1-amine hydrochloride (**4b**)

The starting compound was (3,4-dichlorophenyl)(3-methyl-1,2,3,6-tetrahydroazepino[4,5-b]indol-4-yl)methanone **3b**. The yield of heteroarylmethylated tryptamine derivative **4b** was 89%. Colorless crystals with mp 218–220 °C (from PrOH). ^1^H NMR (600 MHz), DMSO-*d*_6_. δ, ppm: 2.47 (s, 3H, NMe), 2.98 (s, 4H, 2CH_2_), 4.57 (s, 2H, ArCH_2_Het), 6.91–7.01 (m, 2H, H-5′, H-6′), 7.16 (d, *J* 7.9, 1H, H-7′), 7.52 (d, *J* 7.7, 1H, H-4′), 7.68 (dd, *J* 8.3, 2.0, 1H, H-3″), 7.72 (d, *J* 8.3, 1H, H-2″), 7.80–7.85 (m, 2H, H-6, H-7), 7.88 (d, *J* 2.0, 1H, H-6″), 7.98–8.05 (m, 1H, H-8), 8.08–8.15 (m, 1H, H-5), 8.94–8.96 (m, 2H, ^+^NH_2_), 10.57 (s, 1H, NH). ^13^C NMR (150 MHz), DMSO-*d*_6_. δ, ppm: 20.59, 32.22, 33.11, 48.77, 107.15 110.91, 117.70, 118.48, 120.77, 127.45, 128.41, 128.88, 129.31, 130.14, 130.47, 130.53, 130.87, 131.13, 131.85, 132.70, 135.75, 138.85, 140.24, 140.79, 152.38, 152.87. Found (%): C, 62.52; H, 4.31; Cl, 21.14; N, 11.02. Calc. for C_26_H_23_Cl_3_N_4_ (%): C, 62.73; H, 4.66; Cl, 21.36; N, 11.25.

### 4.2. Experimental Animals

The experiments were carried out on 198 white outbred male mice weighing 18–22 g, obtained from the «Rappolovo» nursery (Leningrad region, Russia) and divided randomly into equal groups (*n* = 6). The experiments were approved by the local ethical committee of Volgograd State Medical University, Volgograd, Russia, protocol number: IRB 00005839 IORG 0004900 (OHRP).

### 4.3. Drugs and Treatment

The reference drug diazepam (Sibazon^TM^, J.S.C. Organica, Russia) was used at a dosage of 1 mg/kg. Quinoxaline derivatives **2a**–**4b** were synthesized by the Research Institute of Physical and Organic Chemistry, Southern Federal University, Rostov-on-Don, Russia, and by Federal State Budgetary Scientific Institution “North Caucasian Zonal Research Veterinary Institute”, Novocherkassk, Russia, and were used in screening doses of equimolar to diazepam: 1.73 mg/kg for **2a**, 1.78 mg/kg for **2b**, 1.92 mg/kg for **2c**, 1.98 mg/kg for **2d**, 2.01 mg/kg for **2e**, 1.46 mg/kg for **2f**, 1.75 mg/kg for **2g**, 1.65 mg/kg for **4a** and 1.75 mg/kg for **4b**. The control groups received an equivalent volume of vehicle (distilled water) in a volume of 10 mL/kg of animal`s weight. All compounds were administered orally 30 min before the tests.

### 4.4. Experimental Design

The synthesis of compounds **2a**, **2c**, **2e**, **4a** has been described previously. Compounds **2b**, **2d**, **2f**, **2g**, **4b** were synthesized within this study. The search for new compounds with anxiolytic activity was carried out in vivo using an elevated plus maze test. For an additional assessment of anti-anxiety properties, as well as the formation of a profile of spontaneous motor activity of animals, the open field test was performed. The light-dark box technique was performed for the most active substances to confirm their anxiolytic properties. For the novel series of substances, a pharmacophore feature prediction was conducted to assess the contribution of the chemical structure of compounds to their level of activity. Then, an ADMET investigation was carried out for the most promising substance according to in vivo assay. Acute oral toxicity for that compound has also been tested in vivo.

### 4.5. Behavioral Tests

#### 4.5.1. Elevated Plus Maze

The installation contained two open (36 × 6 cm) and two closed arms (36 × 6 × 15 cm) connected by a central platform (6 × 6 cm) and elevated 60 cm above the floor [42]. The mouse was placed on the platform, facing the open arm, and the percentage (%) of time spent by the animal in the light arm and the number of entries into the open arm were recorded for 5 min. After each animal, the installation was treated with an alcohol solution to remove any odor.

#### 4.5.2. Light/Dark Box

The installation consisted of a chamber 21 × 42 × 25 cm, divided into two equal sections by a partition with a door [43]. One of the sections is brightly lit (390 lux), the other is dark. The mouse was placed in the light section of the setup and monitored via a webcam for 5 min. The percentage (%) of the time spent by the animals in the light part of the set was recorded [44]. A high level of anxiety for the animal was assumed with less than 50% of the time spent in the light chamber [45]. After each animal, the installation was treated with an alcohol solution to remove any odor.

#### 4.5.3. Open Field

The open Field installation was a circular arena with a radius of 40 cm and a side height of 30 cm. The central area of the arena was illuminated with white fluorescent light, there were black lines on the floor with holes at the intersections [17]. The mice were individually placed on the central platform and monitored for 5 min each. The main parameters were recorded, such as locomotor activity and rearings score, search activity and the number of exits to the illuminated center of the installation. Locomotor activity was assessed by the number of quadrants crossed by the animals [46], search activity–by the number of holes examined during the observation period. The proposed test also allows us to give a general assessment of the animal’s behavior. After each animal, the installation was treated with an alcohol solution to remove any odor.

### 4.6. Statistical Analysis of In Vivo Experiments

The statistical processing of the obtained data in vivo was carried out using the Kolmogorov–Smirnov test with the Wilcoxon–Lilliefors test. In the case of normal distribution of the data, one-way ANOVA was used with Dunnett’s post-hoc test; for non-parametric statistics, the Kruskal–Wallis test and Dunn’s post-hoc test were used, implemented in the GraphPad Prism v.8.0 program. Data are presented in the format mean ± mean error (M ± SEM). The *p* < 0.05 values were considered significant.

### 4.7. In Silico

#### 4.7.1. Input Query

The input data set contained 9 novel compounds of quinoxaline derivatives, with various salt components and Diazepam as a reference drug. Chemical structures were built in the ChemOffice in the CDX format. RPBS Web Portal using Frog2 service was used to convert CDX into Mol2 format. All 9 quinoxaline molecules were used as an input query on the PharmaGist webserver to generate the best ligand-based pharmacophore model where Diazepam and NBXQ were set up as a key structure and excluded from data set to evaluate the pharmacophore model of quinoxalines.

##### Pharmacophore Feature Prediction

As the general model was based on diazepam, the construction and search for features of the pharmacophore responsible for the activity of a number of quinoxalines was carried out relative to it. The main assumption was that, based on the structure of diazepam as a reference drug, to identify similar configurations and thus make a general model of pharmacophores. The main PharmaGist scores are shown in Table 6. Some structures have similar chemical formulas but different salt components, and this web service cannot capture subtle differences, that is why overall scores for some compound pairs. In fact, the **4a** and **4b** compound is the closest in characteristics to the reference drug.

The PharmGist algorithm consists of multiple flexible alignments of drug-like substances. Default settings allowed the algorithm iteratively to select each input ligand as a reference. In this case, diazepam played the reference role. After pairwise alignments, the algorithm proceeds to multiple alignments, then to clustering and outputting steps.

The PharmGist algorithm consists of multiple flexible alignments of drug-like substances. By default, the algorithm iteratively selects each input ligand as a reference. In this case, diazepam became the reference point. After pairwise alignments, the algorithm proceeds to multiple alignments, then to clustering and outputting results.

At the first stage, rigid groups are determined for each ligand, as well as a set of physicochemical properties.

The second stage is a pairwise alignment of ligands, a reference ligand and one target ligand. In this case, the target ligand is assumed to be flexible, and the axis is assumed to be rigid. Pairwise comparison creates the highest-scoring K pairwise alignments, where the pairwise alignment is the superposition of the possible conformation of the target ligand on the axis. In the third step, the inputs are a reference ligand, a set of M target ligands, and a large number (K) of pairwise alignments. In this case, similarly to the previous step, the highest score is calculated, only for a set of multiple alignments. Multiple alignment evaluation
(*m +* 1)^1/2^ · *feature_ score*, where feature_score is ∑_i_s(f_i_)

In this case, the algorithm begins with subsets of one matching summary object and gradually expands them until larger subsets no longer match.

When generating pharmacophores, a dataset of multiple alignments between a reference point and various subsets of target ligands is evaluated. In this case, PharmaGist uses the union of the parent multiple alignments with other daughter multiple alignments, for which, due to additional target ligands, the consensus partially coincides with the consensus of the parent. A pharmacophore derived from such an ensemble of multiple alignments is called a weighted pharmacophore. The terminal stage is iterative clustering, for the resulting data set with a change in the reference point.

The main advantage of this method is that in the worst case, when there is no information concerning the binding conformation of any of the ligands, the set of conformations for only one of them (the *pivot*) is calculated [47].

#### 4.7.2. ADMET

At present, calculation of the ADMET properties is an essential part of drug discovery. In this study, we predicted main administration; delivery and toxicology properties of leader compound to detalize further research process and find weaknesses of it.

To predict ADMET we used open access online in silico tools: admetSAR [48], SwissADME [49], ProTox-II [50], pkCSM [51] and DataWarrior [52]. After that, we made a consensus of predicted values to describe compound properties.

### 4.8. Acute Oral Toxicity of Leader Compounds

To validate the results of ADMET calculation, the acute oral toxicity of leader compound **2b** was assessed in mice according to the procedures outlined by the Organization for Economic Co-operation and Development (OECD) [53]. Outbred white mice of either sex were used for the acute oral toxicity studies. Each group consisted of six animals. An additional three male mice and three female mice were given distilled water and were treated as control groups. After the fasting period, the body weight of the mice was determined and the dose was calculated based on the body weight, as the volume of solution administered to the mice is 10 mL/kg. The studied compound **2b** was administered to the animals once, orally, with the aid of a feeding needle connected to syringe [54]. The initial doses were 1000, 1200, 1650, 1900 and 2100 mg/kg. The mice were given food approximately one hour after treatment. Mice were observed in detail for any signs of toxic effect during the first six hours after the treatment period and then daily for 7 days. The LD_50_ dose was calculated for compound **2b**.

## 5. Conclusions

In the course of the study, new derivatives of a C^2^,C^3^-quinoxaline scaffold with supposed psychotropic, in particular anxiolytic, activity were synthesized. According to the results of testing substances in vivo, the compounds under codes **2a**, **2b**, **2c** and **4b** showed activity at the level of the comparison drug diazepam. Compound **2b,** in terms of the light/dark box technique, showed an effect even higher than diazepam. Prediction results revealed that the greatest accuracy in identifying active compounds was in the group without diazepam and NBQX. The in silico prediction showed the promise of compound **2b** in searching for the activity of compounds of similar chemical groups by the pharmacophore feature method. For the selected compound, a class IV of acute toxicity was determined. Thus, further investigating the properties of **2b** is promising due to its low toxicity and high anxiolytic effect in comparison to the reference drug diazepam.

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
