# Peer review of "Design, Synthesis and Pharmacological Evaluation of Novel C2,C3-Quinoxaline Derivatives as Promising Anxiolytic Agents"

_ijms, 2022, doi:10.3390/ijms232214401_

Round 1

Reviewer 1 Report

This paper reports the synthesis and behavioral pharmacological testing of several new derivatives of C2,C3-quinoxaline.  Nine new compounds were evaluated in the elevated plus maze and open field tests, while four compounds were evaluated a light/dark box test.  In silico computational work was performed to predict potential key pharmacophore features of active compounds.  Predicted ADMET properties were also derived in silico for lead compound 2b, and finally acute oral toxicity was measured for compound 2b.

The main strength of this paper is the behavioral pharmacology presented.  The paper demonstrates that some derivatives of C2,C3-quinoxaline are active in behavioral assessments of anxiety.  This data is key in suggesting development of these compounds as anxiolytic agents may be possible.

The in silico results provide less convincing data. The predicted pharmacophore features provide a glimpse into possible structural elements that are key to the compounds function. However, there is a lack of discussion of potential mechanism of action for these compounds, and there is no discussion of potential structure-activity relationships between analogues with distinct behavioral affects.  The introduction mentions quinoxalines acting as 5-HT3 receptor antagonists, but no suggestion is put forth for the compounds tested in this paper.  In addition, The ADMET results for compound 2b would be greatly strengthened with empirical results over in silico predictions.  These ADMET properties could be test in vitro along with pharmacokinetic values for plasma and brain for the lead compound.  This would provide more substantial information about the potential use and liabilities of compound 2b for therapeutic purposes. 

Specific comments 

Line 88:  Table 1

The general structure on the left has no R present.  Please correct so the R group is labeled on the structure.

Line 99:  Significant Digits

Make significant digits similar for the mean and the SEM. For example, if mean is 6.8 then only report SEM of 1.2%

Line 102:  Figure 1

Please provide the p values for the different “*” noted in the Figure for both panel A and B.  Three different significance stars are used in the figure *, ** and ****.

Line 204:  Figure 6

Please provide the actual doses (mg/kg) tested in the figure legend.  The methods on line 474 suggest doses of 5, 50, 300, and 2000 mg/kg, but that does not match the doses in the Figure.

Line 359

Please provide actual doses (mg/kg) of each compound administered.

Author Response

Dear editor! Thank you very much for your comments. It helps us to improve our submissions in future. We are going to continue the investigation on the mechanisms of action of the leader compound in our follow-up works. Also we accepted all of the specific comments you noted. Once again we are very grateful for the work gone with our article.

Reviewer 2 Report

In the manuscript entitled "Design, synthesis and pharmacological evaluation of novel C2,C3-quinoxaline derivatives as promising anxiolytic agents", the authors design and synthesize quinoxaline derivatives, and compound 2b exhibits anti-anxiety activity in behavioral tests. However, I do not think the results are enough to be published in IJMS.

1. Although the authors demonstrate the anti-anxiety activity of the derivatives in behavioral tests, their potential mechanisms in neurotransmitters and signal transduction are not involved.

2. The test results are not prominent in the Abstract and some Keywords are redundant.

3. The data of high resolution mass spectrometry of target compounds need to be added.

4. In Scheme 1 and Table 1, the labeling of compound substituents is confusing, 1,2-diaminobenzene in the two synthetic routes is inconsistent, and some groups in the derivative structures are not standardized, such as -CH2CR2NHMe, -CH2CH2Me, -CMe3.

5. The dosage of derivatives administration is not mentioned in behavioral tests.

6. Data in Figure 1 and Figure 2 have large errors. Why did the results show no significant difference in Figures 2B, 2D and 2E?

7. Molecules in Table 2-4 are confusing.

8. Compound numbers are not bold in Discussion.

Author Response

Dear editor! Thank you very much for your comments. It helps us to improve our submissions in future. 

  1. We are going to continue the investigation on the mechanisms of action of the leader compound in our follow-up works.
  2. Corrected.
  3. This article describes the anxiolytic activity of [3-(hetero)arylquinoxalin-2-yl]methyl derivatives of β-(hetero)arylethylamines 2a-g, 4a, 4b. Some of these compounds were described by us earlier [32], where the structure of the ancestor of this series of quinoxaline derivatives, compound 2a, also studied in this article, was proved using X-ray diffraction analysis. All other derivatives of this system were obtained by a similar procedure, and the data of elemental analysis and 1H- and 13C NMR spectroscopy convincingly prove the structure of the obtained compounds.
  4. Corrected
  5. Corrected
  6. In psychopharmacological tests the wide range of animal`s reactions is often happens. We used 6 animals in each experimental group. Thus, in some of parameters of open field there were no significant difference between groups. At the same time, in our opinion, these data are the essential part of the investigation, as the negative result is also the result of the work.
  7. Corrected to
  8. Corrected

Once again we are very grateful for the work gone with our article.

Reviewer 3 Report

The authors describe the synthesis and pharmacological evaluation of quinoxaline derivatives 2a-4b as promising anxiolytic agents. However, with minor revisions, this review would be reasonable to accept for publication in the International Journal of Molecular Sciences.

1.     I Suggest authors to move general information (2.1 Chemistry line 77-87) to the Materials and Methods section.

2.     Replace CMe3 with tBu throughout the manuscript.

3.     In page no 10 Line 268 correct the dried with anhydrous Na2CO3.

(Authors may use anhydrous Na2SO4)

4.     In page no 10 Line 271 Replace and EtOH with in EtOH.

5.     Provide the solvents for recrystallization of 2b, 2d, 2f, 2g, 4b in the experimental data.

6.     In page no 10 Line 277 Replace 1a with 1b.

7.     In page no 11 Line 292 Replace [32] with 1d.

8.     In page no 11 Line 309 Replace 1a with 1f.

9.     In page no 12 Line 309 starting material number as 3b.

10.  In page no 10 Line 279 Replace 1H NMR with 1H NMR.

Author Response

Dear editor! Thank you very much for your comments. It helps us to improve our submissions in future. We would like to keep point 2.1 Chemistry as it is because this briefly describes the basics of our work and then goes into more detail in the Materials and Methods section. All your other comments have been accepted and mistakes fixed. 

Once again we are very grateful for the work done with our article.

Reviewer 4 Report

The authors in this manuscript wanted to explore the anxiolytic property of the quinoxaline derivatives listed in the manuscript, with the view that quinoxaline has shown anxiolytic properties in previous precedence. The compound 2b was found to have the most anxiolytic property as found in their mouse behavioral tests that are pharmacologically validated experimental models. For most part, I agree with the findings in the paper and the methods employed.

There are a few issues in their schemes for the structures of the compounds. The numbering descriptors in Scheme 1 are confusing and can definitely be improved. Also the structures in the Table 1 all show a dashed stereo center, which is not correct as that bond connects to sp2 carbons. That definitely needs to be rectified. 

Further, upon analysis of the results and its impact I believe that the findings merit publication, but does not meet the standards for this journal.

Author Response

Dear editor! Thank you very much for your comments. It helps us to improve our submissions in future. We improved the chemical part of the article and hope that our submission became more suitable now. 

Round 2

Reviewer 2 Report

Compared with the previous version of the manuscript, the experimental content has not been improved.

Author Response

Dear editor
We are embarrassed by your comment, because last time we tried to take into account all your comments when correcting
Could you please elaborate on the points of your wishes so that we can correct our mistakes?
Thank you very much for your time

Reviewer 4 Report

The manuscript did satisfactory changes to the schemes and tables in the chemical information part in Table 1 and scheme 1. The stereochemistry of the dashed bonds were replaced with arrows showing the site of attachment which is good. But then in the note it said site of attachment of the radical, which opens a can of worms about the mechanism of the reaction which certainly does not proceed through a radical pathway. I would omit the word "radical" and just write site of attachment to the quinoxaline moiety.

Also no change was made to the portion I highlighted were the use of the word "relatively" was ungrammatical. It should be chemical shifts of those nuclei were measured "relative to the reference peaks" of deuterated solvents.

Author Response

Dear Editor
We tried to take into account all your corrections and hope that we succeeded.
Thanks a lot for your time